# An ECG Classification Method Based on Multi-Task Learning and CoT Attention Mechanism

**DOI:** 10.3390/healthcare11071000

**Published:** 2023-03-31

**Authors:** Quancheng Geng, Hui Liu, Tianlei Gao, Rensong Liu, Chao Chen, Qing Zhu, Minglei Shu

**Affiliations:** 1Shandong Artificial Intelligence Institute, Qilu University of Technology (Shandong Academy of Sciences), Jinan 250014, China; 2Department of Cardiology, Qilu Hospital of Shandong University, Jinan 250012, China

**Keywords:** ECG, SE-ResNet, multi-task deep neural network, Contextual Transformer

## Abstract

Electrocardiogram (ECG) is an efficient and simple method for the diagnosis of cardiovascular diseases and has been widely used in clinical practice. Because of the shortage of professional cardiologists and the popularity of electrocardiograms, accurate and efficient arrhythmia detection has become a hot research topic. In this paper, we propose a new multi-task deep neural network, which includes a shared low-level feature extraction module (i.e., SE-ResNet) and a task-specific classification module. Contextual Transformer (CoT) block is introduced in the classification module to dynamically model the local and global information of ECG feature sequence. The proposed method was evaluated on public CPSC2018 and PTB-XL datasets and achieved an average F1 score of 0.827 on the CPSC2018 dataset and an average F1 score of 0.833 on the PTB-XL dataset.

## 1. Introduction

Cardiovascular disease is responsible for the deaths of approximately 17.9 million people per year, accounting for 31% of all global fatalities [1]. Cardiovascular disease has become the disease with the highest mortality rate and has seriously threatened human life and health. Electrocardiogram (ECG) is the most widely used noninvasive heart disease diagnosis technology. The ECG signal represents electrical changes during the cardiac cycle and can be recorded with the surface electrodes. Analysis of ECG [2,3,4] signals allows doctors to gather important insights into the health condition of patients and to promptly identify heart abnormalities, thereby prolonging life and improving quality of life through appropriate treatment.

With the rapid advancement of mobile devices for ECG monitoring, automatic interpretation of ECG recordings is becoming more and more important for the early detection and diagnosis of arrhythmias [5]. In particular, many effective arrhythmia classification methods for single-lead ECG have been proposed [6,7,8], but single-lead ECG alone is not enough to accurately diagnose various heart diseases. 12-lead ECG can comprehensively evaluate cardiac electrical activity (including arrhythmia and myocardial infarction) and each lead reflects heart states through electrical signal changes from different perspectives. Therefore, as a standard clinical ECG examination, 12-lead ECG has attracted more and more interest from researchers [9,10,11,12,13].

Over the past few years, various machine learning methods have been employed to analyze ECG signals, such as decision tree [14], support vector machine [15] and hidden Markov model [16]. The most critical part of these methods is to extract discriminant information from original ECG data, i.e., feature extraction. In order to extract features, many methods have been proposed, which can be divided into two categories: manual approaches [17,18] and automatic approaches [19,20,21]. The manual approaches mainly rely on professional medical knowledge and the rich expertise of cardiologists. For example, Martis et al. extracted features such as RR interval, R peak amplitude and QRS duration from the original ECG data and used decision trees to diagnose arrhythmia types [21]. In addition, wavelet transform [22], short-time Fourier transform and principal component analysis [23] are often used to extract time-frequency features. It is difficult to obtain satisfying performance for manual approaches because ECG features such as PR interval show a large diversity for different individuals.

Deep learning [24,25,26,27] is an efficient feature learning method that employs neural networks for mining useful features from a large amount of data. Due to its ability to effectively express unstructured data with non-linear representation, deep learning has been successfully applied in a variety of fields including computer vision, speech recognition and natural language processing [28,29,30]. Due to its powerful ability, many researchers have used deep learning-based methods for ECG classification [31,32,33,34], which show higher accuracy than traditional methods. For example, Rahhal et al. [29] proposed to use continuous wavelet transform to convert one-dimensional ECG signals into two-dimensional images and then used convolutional neural network (CNN) to extract useful features of images, yielding a high accuracy for identifying abnormal ventricular beats. Hannun et al. [34] built a 34-layer CNN network and obtained a cardiologist-level F1 score of 0.837 for classification of twelve arrhythmias. Other common deep learning models that have been used for ECG classification include Long-Short Term Memory (LSTM) network [35], Recurrent Neural Network (RNN) [36] and bidirectional LSTM network [37].

Although deep learning methods have achieved promising performance in ECG classification, they still have some problems. The feature map derived from different leads of the 12-lead ECG signal may exhibit varying levels of contributions for detecting arrhythmias. Specifically, the spatial characteristics of arrhythmias, such as the morphology of waveform, may differ between leads. For example, atrial fibrillation is most obvious in leads II and V1 [38]. In addition, previous studies mainly focus on one ECG classification task, though there are many kinds of ECG tasks (e.g., anomaly detection, heartbeat classification, ECG delineation and ECG diagnosis) in the real world and there exist correlations between tasks, which indicates that the combination of different tasks is conducive to learning better ECG representations [39].

Multi-task learning has achieved great success and has performed well in many fields, such as natural language processing [40], speech recognition [41], computer vision [42] and face verification [43]. In this paper, we propose a deep multi-task learning [44] method for ECG classification. Specifically, we first construct a related auxiliary classification task with a corresponding dataset by merging similar classes (e.g., left bundle branch block and right bundle branch block) or splitting large classes. We propose a multi-task deep neural network, which includes a shared low-level feature extraction module (i.e., SE-ResNet) and a task-specific classification module. The main contributions of this study are as follows:A new multi-task network consisting of a shared low-level feature extraction module and a task-specific classification module is proposed for ECG classification.We propose two strategies to create auxiliary tasks, which exploit the hierarchical class information to achieve feature sharing between the main task and the auxiliary task.Contextual Transformer (CoT) block is first introduced in ECG classification to solve the problem that the self-attention mechanism only focuses on the local information of the ECG sequence.

## 2. Materials and Methods

### 2.1. Data

#### 2.1.1. CPSC2018 Dataset

The 2018 China Physiological Signal Challenge (CPSC2018) dataset [45] includes 6877 publicly accessible 12-lead ECG records (female: 3178; male: 3699) and a private test set consisting of 2954 12-lead ECG records. ECG records were collected from 11 hospitals. Record lengths range from 6 s to 60 s with a sampling rate of 500 Hz. Table 1 shows the number of records for each class and superclass. In addition, we also show the specific description of each class in Table 1.

#### 2.1.2. PTB-XL Dataset

The PTB-XL dataset [46] includes 21,837 10-second 12-lead clinical ECG records from 18,885 patients, 52% of whom were male and 48% female. The sampling frequency is 500 Hz for all records. There are three levels of ECG labels in the dataset: 5 superclasses, 23 subclasses and 44 diagnostic statements. Table 2 shows the number of records for each superclass and subclass. In this study, we used the recommended data partition scheme of training and test sets [46]. The whole data set was split into 10 folds and the ninth and tenth folds contain only ECGs that were validated by at least one cardiologist, which were recommended to be used as validation and test sets, respectively. The remaining eight folds were used for training.

### 2.2. Preprocessing

In this study, data preprocessing is performed according to previous work [47]. Specifically, ECG signals are downsampled from 500 Hz to 250 Hz to accelerate the training. In addition, signals in the CPSC2018 dataset do not have the same length; therefore, they are cropped or padded to 60 s with zero since the convolutional neural network cannot accept input of different lengths in a batch.

### 2.3. Creating Auxiliary Task

Previous studies mostly focus on a single task and ignore other potentially helpful information. This study proposes a multi-task model to simultaneously learn two tasks related to ECG classification, in which the feature representation learned from one task can also be utilized for the other task. Compared with single-task model, the multi-task model has multiple independent outputs on their own paths, which helps to prevent overfitting and enhances the model’s generalizability.

In this study, we construct auxiliary tasks based on the relationships between ECG classes. For the CPSC2018 dataset, the classification of the original nine classes is the main task. Then we reorganize the original nine classes into five super classes (see Table 1), which are used for the auxiliary task. For the PTB-XL dataset, the main task is the classification of five classes. Then we reuse the 23 subclass-level labels (see Table 2) as the target of auxiliary task.

### 2.4. Multi-Task Neural Network

We design a multi-task deep neural network using hard parameter sharing. As shown in Figure 1, it mainly includes a shared neural network (SE-ResNet) and feature extraction networks for specific tasks. By using a shared network structure prior to the multi-task module, the model is able to learn both tasks concurrently using a shared representation.

#### 2.4.1. Squeeze-Excitation-ResNet (SE-ResNet)

We propose the SE-ResNet as the shared network by both tasks. Inspired by ResNet [48] design, SE-ResNet is composed of multiple SE blocks. As shown in Figure 2a, each SE block consists of two convolutional layers with a kernel size of 7. A batch normalization (BN) layer is applied after each convolutional layer and we use the rectified linear unit (ReLU) activation function [49] as the nonlinear transformation function. BN helps to speed up the convergence of the model during training by normalizing the data in each batch. A dropout rate of 0.2 is set to prevent overfitting of the neural network.

The last BN layer is followed by an SE module (see Figure 2b). The SE module can obtain the importance of feature information to strengthen the guiding role of channel information in the ECG classification process. The SE module includes two parts: squeeze and excitation. The squeeze operation seeks to use a value with a global receptive field to reflect the significance of each channel feature and to generate a feature map through global mean pooling of each channel. The excitation operation uses the fully connected layer to act on the feature map and the importance weight of each channel is applied to the corresponding channel to construct the correlation between the channels.

#### 2.4.2. Contextual Transformer (CoT) Block

The traditional self-attention [50] mechanism characterizes interactions between different positions of feature sequence well, only depending on the input itself. However, in the traditional self-attention mechanism, all pairs of query keyword relationships are learned independently on isolated query keyword pairs without exploring the rich context between them, which severely limits the ability for self-attention learning on ECG sequences. To alleviate this problem, we employ a new Contextual Transformer (CoT) block [51] in this study, which integrates contextual information mining and self-attention learning into a unified architecture. The CoT attention mechanism promotes self-attention learning by using additional contextual information between input keys and ultimately improves the representation characteristics of deep networks. The structure of the CoT block is shown in Figure 3.

We define Q=E, K=E, V=EW for the ECG signal sequence *E* extracted by SE-ResNet, where *Q* and *K* are the values of the original ECG signal sequence, *V* is the value of the feature mapping of the ECG signal and *W* represents an embedding matrix. Firstly, we use 3 × 3 convolution to statically model *K* to obtain with local sequence information representation. Secondly, in order to interact query information with local sequence information $K_{1}$, we need to concatenate $K_{1}$ and *Q* and then generate an attention map after continuous convolution operation. The formula is as follows:(1)$$A=\left[K_{1},Q\right]W_{\sigma }W_{\theta }$$

We then need to multiply *A* and *V* to get the sequence $K_{2}$ with global sequence information by:(2)$$K_{2}=V*A$$

Finally, we concatenate the local sequence information $K_{1}$ and the global sequence information $K_{2}$ to obtain the final output result *D*.
(3)$$D=[K_{1},K_{2}]$$

#### 2.4.3. Bi-GRU Module

The GRU [52] layer can model the relationship of long-term context, but one drawback is that it can only read the input ECG signal sequence information from a single direction. In this study, we use the Bi-GRU layer to capture the characteristics of the input ECG feature map from the positive and negative directions, respectively.

We will get the matrix D through the CoT attention layer as the input of Bi-GRU. In the forward layer of the Bi-GRU layer, the input vectors (from $x_{t}$ to $x_{t+1}$) are read in a positive order and the forward hidden layer state corresponding to each vector is calculated, namely ($\vec{h_{1}}$, …, $\vec{h_{t}}$, …, $\vec{h_{T}}$). Similarly, in the backward layer of the Bi-GRU layer, the input vectors (from $x_{t}$ to $x_{t-1}$) are read in reverse order and the reverse hidden layer state corresponding to each vector is calculated. The structure of the Bi-GRU layer is shown in Figure 4.

### 2.5. Loss Function

Feature vectors output by the Bi-GRU layer are sent to the last fully connected layer, which generates the class distribution of the two tasks. For two tasks, we calculate the loss using the cross entropy function. Then two loss values are added with different weights. The final loss is as follows:(4)$$L^{\left(total\right)}=\lambda L^{\left(main\right)}+\left(1-\lambda \right)L^{\left(aux\right)}$$
where λ is the weighting parameter, $L^{main}$, $L^{aux}$ represent the loss of the main task and auxiliary task, respectively.

## 3. Experimental Setup

### 3.1. Parameter Setting

For both datasets, we use the Adam optimizer and set the initial learning rate to 0.0005. The learning rate is reduced by a factor of 10 every 10 epochs. We set the batch size to 32. Ten-fold cross-validation is used in this work, where the validation set is used for model and parameter selection and the test set is used to test the effectiveness of the network. In order to alleviate the over-fitting during the training process, we also adopted an early stop mechanism, i.e., we will end the training when the loss is not decreasing for 10 consecutive epochs.

### 3.2. Evaluation Metrics

In this study, the F1 score is used to measure the model classification performance for each class. It is a harmonic mean of precision and recall, defined as follows:(5)$$\mathrm{Precision}=\frac{\mathrm{TP}}{\mathrm{TP}+\mathrm{FP}}$$
(6)$$\mathrm{Recall}=\frac{\mathrm{TP}}{\mathrm{TP}+\mathrm{FN}}$$
(7)$$F1=\frac{2\times (\mathrm{Precision}\times Recall)}{\mathrm{Precision}+\mathrm{Recall}}$$

The terms TP, FP and FN refer to the number of true positive, false positive and false negative samples, respectively. For the overall evaluation, we use the macro F1 score, i.e., the arithmetic mean of F1 scores of all classes and the area under the ROC curve (AUC).

## 4. Experiment and Results

### 4.1. Results on the PTB-XL Dataset

The results are reported in Table 3. The average F1 score reaches 0.833 and the average AUC value reaches 0.925. The precision values of all classes are larger than 0.810 and the recall values of all classes except the MI are also larger than 0.810. This is because the MI superclass mainly includes two types of subclasses: AMI and IMI. AMI is mainly caused by V1-V6 leads and IMI is mainly caused by II,III,aVF. The difference between the two leads affects the judgment of our model. Among all classes, the model performs best for CD and HYP with AUC scores exceeding 0.930. Figure 5 shows the overall classification performance of this method and seven previous studies, which are all single-task classification. The highest F1 score produced by other studies reaches 0.823, which is lower than 0.833.

### 4.2. Results on the CPSC2018 Dataset

The results on the CPSC2018 dataset are reported in Table 4. Overall, the proposed method yields an average AUC value of 0.977, an average accuracy of 0.966, an average F1 score of 0.827, an average precision of 0.852 and an average recall of 0.808. Among all arrhythmias, the model performs best for LBBB and RBBB with F1 scores exceeding 0.930.

Table 5 shows per-class and overall classification performance of this method and five previous studies. The highest F1 score produced by other studies reaches 0.813, which is lower than 0.827. Our model is superior to other models in the diagnosis of SNR, IAVB, LBBB, RBBB, STD and STE.

### 4.3. Effect of Random Auxiliary Task

We use two strategies when creating auxiliary tasks. The first is to merge similar classes into superclasses (i.e., CPSC2018 dataset) and the second is to split one class into more subclasses (i.e., PTB-XL dataset). Obviously, the classes of auxiliary task are still organized hierarchically.

In order to verify the effectiveness of the proposed strategies, we conduct the experiments with random auxiliary tasks on the CPSC2018 dataset and the PTB-XL dataset, respectively, e.g., we randomly merge classes into superclasses which do not make sense from a clinic perspective. As shown in Table 6 and Table 7, the average F1 score for CPSC2018 and PTB-XL datasets reaches 0.807 and 0.818, respectively, which is comparable to that produced by previous single-task classification and is still lower than that produced with proposed auxiliary task.

### 4.4. Ablation Experiment

In order to prove the effectiveness of the CoT attention mechanism and the Bi-GRU layer, we conduct ablation experiments about these two modules. Results produced on the PTB-XL and CPSC2018 datasets by our model without the CoT block are shown in Table 8 and Table 9. It can be observed that metrics have decreased overall. The F1 score decreases from 0.833 to 0.819 on the PTB-XL dataset and from 0.827 to 0.761 on the CPSC2018 dataset.

Results produced on the PTB-XL and CPSC2018 datasets by our model without the Bi-GRU layer are shown in Table 10 and Table 11. It can be also observed that metrics decrease overall. The F1 score decreases from 0.833 to 0.820 on the PTB-XL dataset and decreases from 0.827 to 0.807 on the CPSC2018 dataset, though the accuracy decrease is not as much as that produced by removing the CoT block.

## 5. Discussion

Experimental results demonstrate the effectiveness of our proposed method. As shown in Table 5 and Figure 5, our method produced an overall F1 score of 0.833 and 0.827 on the PTB-XL and CPSC2018 datasets, respectively, which are higher than the optimal F1 scores produced by other methods by 0.01 and 0.014. Among different classes of the CPSC2018 dataset, as shown in Table 5, our method shows advantages for SNR, LBBB, STD, STE and is inferior to other methods for AF, PVC, etc. In addition, as shown in Table 3, the F1 score of MI is lower than the other four superclasses. Therefore, we need to try different preprocessing techniques (including different sampling rates or noise reduction methods, etc.) to improve the classification performance of MI in the next work.

We find that the auxiliary task has an impact on the main task. For example, LBBB and RBBB belong to the same superclass in the auxiliary task and our method yields similar scores (0.937 and 0.939) for them, while there is a large lap between the two scores for other methods. When the correlation between the main task and auxiliary task is high, the shared SE-ResNet can be well trained in parallel by two tasks. As shown in Table 6 and Table 7, by contrast, the correlation between the main task and random auxiliary task is relatively small, the advantage of multi-task learning cannot be exhibited, indicating the effectiveness of the proposed strategy for creating auxiliary tasks.

Table 8, Table 9, Table 10 and Table 11 demonstrate the effectiveness of the CoT attention mechanism and the Bi-GRU layer. CoT block can simultaneously extract key features of static local information and global dynamic information from ECG sequences. Although the Bi-GRU module also plays an important role in processing sequence information, the accuracy drop caused by removing the Bi-GRU layer is not as much as that caused by removing the CoT block.

In addition, to make the results more distinct, we use three decimals consistently for all results including F1 scores in this study. Two decimals may lead to indistinguishable comparison, e.g., three methods would have the same F1 score of 0.81 in Table 5. Although a value at the third decimal may be insignificant in the statistical sense, many studies still presented F1 scores with three decimals for better comparison of different methods [11,30,47].

## 6. Conclusions

In this work, a new multi-task deep neural network, which includes a shared low-level feature extraction module (i.e., SE-ResNet) and a task-specific classification module, is proposed. Contextual Transformer (CoT) block is introduced in the classification module to solve the problem that the self-attention mechanism only focuses on the local information of the ECG feature sequence. Thus, the proposed method can dynamically model the local and global information of the ECG feature sequence. In addition, we propose to create auxiliary tasks by merging similar classes or splitting large classes, which exploit the hierarchical class information. The proposed method produced an overall F1 score of 0.833 and 0.827 on the PTB-XL and CPSC2018 datasets, respectively, which are higher than the optimal F1 scores produced by other methods by 0.01 and 0.014, suggesting the effectiveness of our method. However, our multi-task method still has some limitations. Specifically, our model shows bad performance in detecting some abnormal classes, such as detecting the superclass MI on the PTB-XL dataset. In addition, we only use hard parameter sharing to achieve multi-tasking without considering the soft parameter implementation. Finally, we hope that the arrhythmia detection method proposed in this paper can play a role in the process of early diagnosis of cardiovascular diseases. In the future, we will consider integrating more auxiliary tasks for improving the performance of the main task and investigate other approaches to creating auxiliary tasks. With more tasks, we also need to improve the network structure for computational efficiency.

## Figures and Tables

**Figure 1 healthcare-11-01000-f001:**
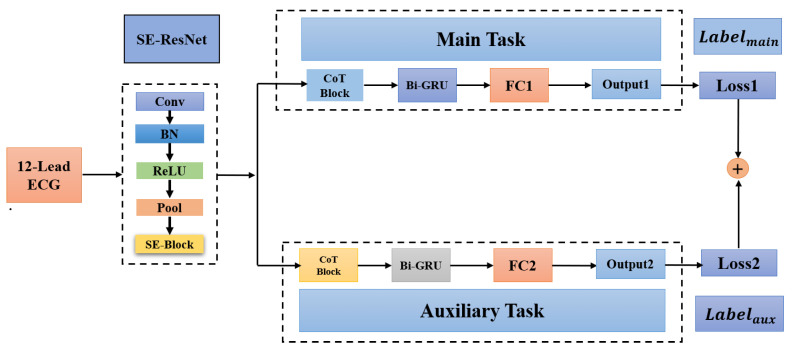
Multi-task neural network. FC1 and FC2 are fully connected layers used in the main task and the auxiliary task, respectively.

**Figure 2 healthcare-11-01000-f002:**
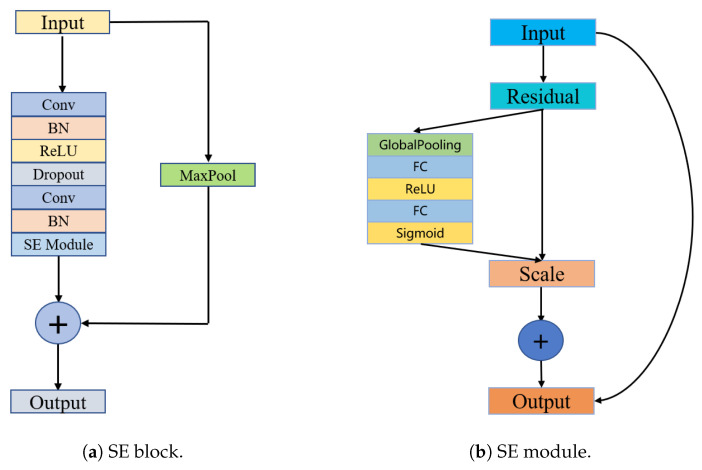
SE-ResNet.

**Figure 3 healthcare-11-01000-f003:**
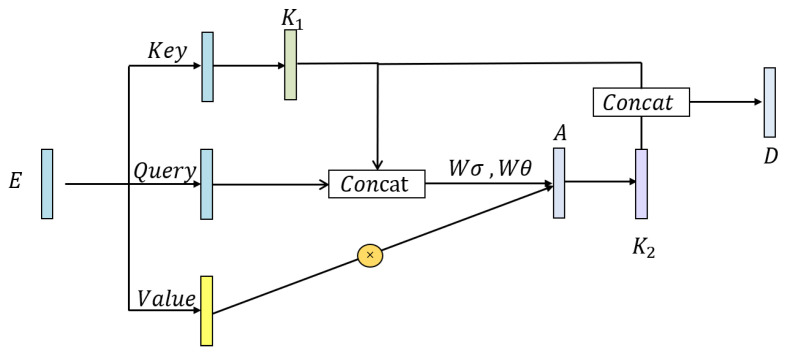
Structure of CoT block.

**Figure 4 healthcare-11-01000-f004:**
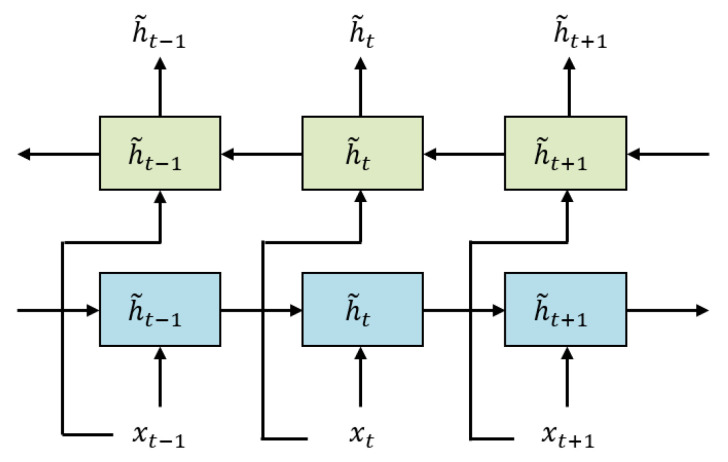
Structure of Bi-GRU layer.

**Figure 5 healthcare-11-01000-f005:**
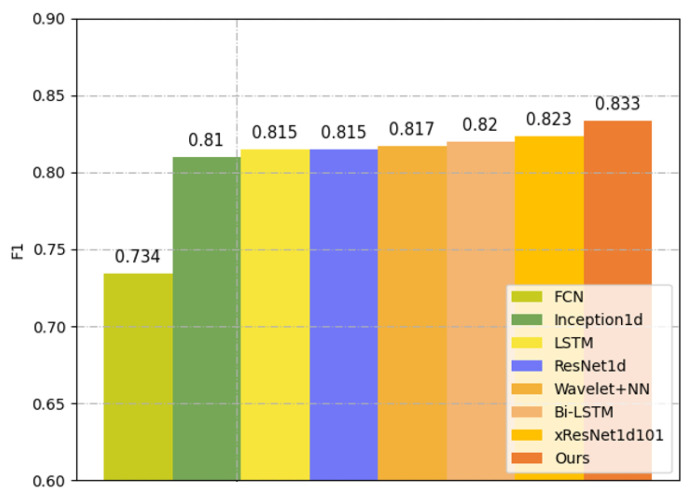
Classification performance of the proposed method and seven other methods (FCN [11], Inception1d [12], LSTM [35], ResNet1d [25], Wavelet+NN [13], Bi-LSTM [37], and xResNet1d101 [26]) on the PTB-XL dataset.

**Table 1 healthcare-11-01000-t001:** ECG classes and their numbers in the CPSC2018 dataset.

Class	Description	Superclass	#Records
SNR	Normal ECG	NORM	918
AF	Atrial fibrillation	AF	1098
IAVB	First-degree atrioventricular block		704
LBBB	Left bundle branch block	QRS	207
RBBB	Right bundle branch block		1695
PAC	Premature atrial contraction	V	556
PVC	Premature ventricular contraction		672
STE	ST-segment elevated	ST	825
STD	ST-segment depression		202
Total			6877

**Table 2 healthcare-11-01000-t002:** ECG classes and their numbers in the PTB-XL dataset.

Superclass	Subclass	Description	#Records
NORM	NORM	Noraml ECG	9528
	LAFB/LPFB	left anterior/posterior fascicular block	1803
	IRBBB	incomplete right bundle branch block	1118
	ILBBB	incomplete left bundle branch block	77
CD	CLBBB	complete left bundle branch block	536
	CRBBB	complete right bundle branch block	542
	_AVB	AV block	827
	IVCD	non-specific intraventricular conduction disturbance	789
	WPW	Wolff–Parkinson–White syndrome	80
	LVH	left ventricular hypertrophy	2137
	RVH	right ventricular hypertrophy	126
HYP	LAO/LAE	left atrial overload/enlargement	427
	RAO/RAE	right atrial overload/enlargement	99
	SEHYP	septal hypertrophy	30
	AMI	anterior myocardial infarction	3172
	IMI	inferior myocardial infarction	3263
MI	LMI	lateral myocardial infarction	201
	PMI	posterior myocardial infarction	17
	ISCA	ischemic in anterior leads	1016
	ISCI	ischemic in inferior leads	398
STTC	ISC_	non-specific ischemi	1275
	STTC	ST-T changes	2329
	NST_	non-specific ST changes	770
Total			21,837 ^†^

† The total number of class labels is 30,560.

**Table 3 healthcare-11-01000-t003:** Classification performance of the proposed method on the PTB-XL dataset.

Type	AUC	Accuracy	Precision	Recall	F1
NORM	0.924	0.905	0.877	0.849	0.863
MI	0.898	0.912	0.814	0.734	0.772
STTC	0.926	0.869	0.834	0.813	0.823
CD	0.946	0.868	0.867	0.872	0.869
HYP	0.933	0.883	0.852	0.819	0.835
AVG	0.925	0.887	0.849	0.817	0.833

**Table 4 healthcare-11-01000-t004:** The classification performance of the proposed method in CPSC2018.

Type	AUC	Accuracy	Precision	Recall	F1
SNR	0.973	0.948	0.824	0.824	0.824
AF	0.981	0.962	0.885	0.968	0.925
IAVB	0.981	0.977	0.923	0.845	0.882
LBBB	0.999	0.996	0.957	0.917	0.937
RBBB	0.994	0.965	0.944	0.934	0.939
PAC	0.961	0.951	0.712	0.758	0.734
PVC	0.962	0.959	0.885	0.676	0.767
STD	0.984	0.962	0.868	0.805	0.835
STE	0.960	0.977	0.667	0.545	0.600
AVG	0.977	0.966	0.852	0.808	0.827

**Table 5 healthcare-11-01000-t005:** Comparison for classification performance of previous works and ours evaluated on the recommended test set of CPSC2018.

Model, Year	F1									
	SNR	AF	IAVB	LBBB	RBBB	PAC	PVC	STD	STE	AVG
CNN + LSTM [30], 2018	0.753	0.900	0.809	0.874	0.922	0.638	0.832	0.762	0.462	0.772
CNN + LSTM [31], 2020	-	-	-	-	-	-	-	-	-	0.806
CNN + Attention [32], 2019	0.790	0.930	0.850	0.860	0.930	0.750	0.850	0.800	0.560	0.813
CNN + LSTM + Attention [33], 2020	0.789	0.920	0.850	0.872	0.933	0.736	0.861	0.789	0.556	0.812
Interpretable ResNet [47], 2021	0.805	0.919	0.864	0.866	0.926	0.735	0.851	0.814	0.535	0.813
Ours	0.824	0.925	0.882	0.937	0.939	0.734	0.767	0.835	0.600	0.827

**Table 6 healthcare-11-01000-t006:** Classification performance with the random auxiliary task on the CPSC2018 dataset.

Type	AUC	Accuracy	Precision	Recall	F1
SNR	0.974	0.952	0.848	0.824	0.836
AF	0.980	0.965	0.866	0.915	0.890
IAVB	0.983	0.975	0.897	0.859	0.878
LBBB	0.999	0.996	0.957	0.917	0.937
RBBB	0.993	0.969	0.936	0.960	0.948
PAC	0.952	0.948	0.671	0.823	0.739
PVC	0.969	0.962	0.809	0.765	0.786
STD	0.980	0.956	0.777	0.890	0.830
STE	0.956	0.968	0.500	0.318	0.389
AVG	0.976	0.966	0.807	0.808	0.807

**Table 7 healthcare-11-01000-t007:** Classification performance with the random auxiliary task on the PTB-XL dataset.

Type	AUC	Accuracy	Precision	Recall	F1
NORM	0.917	0.889	0.845	0.839	0.842
MI	0.883	0.907	0.807	0.699	0.749
STTC	0.923	0.863	0.840	0.781	0.809
CD	0.935	0.859	0.861	0.865	0.863
HYP	0.925	0.879	0.851	0.806	0.828
AVG	0.917	0.879	0.841	0.798	0.818

**Table 8 healthcare-11-01000-t008:** Results produced by the proposed model without the CoT block on the PTB-XL dataset.

Type	AUC	Accuracy	Precision	Recall	F1
NORM	0.908	0.893	0.870	0.815	0.842
MI	0.881	0.904	0.782	0.735	0.758
STTC	0.917	0.865	0.824	0.816	0.820
CD	0.935	0.859	0.858	0.861	0.859
HYP	0.911	0.869	0.827	0.808	0.817
AVG	0.910	0.878	0.832	0.807	0.819

**Table 9 healthcare-11-01000-t009:** Results produced by the proposed model without the CoT block on the CPSC2018 dataset.

Type	AUC	Accuracy	Precision	Recall	F1
SNR	0.973	0.946	0.835	0.794	0.814
AF	0.979	0.958	0.860	0.868	0.864
IAVB	0.979	0.971	0.905	0.803	0.851
LBBB	0.998	0.988	0.750	0.900	0.818
RBBB	0.992	0.964	0.918	0.960	0.939
PAC	0.956	0.948	0.741	0.645	0.690
PVC	0.960	0.961	0.918	0.662	0.769
STD	0.983	0.951	0.809	0.878	0.842
STE	0.941	0.957	0.444	0.182	0.258
AVG	0.973	0.960	0.798	0.744	0.761

**Table 10 healthcare-11-01000-t010:** Results produced by the proposed model without the Bi-GRU module on the PTB-XL dataset.

Type	AUC	Accuracy	Precision	Recall	F1
NORM	0.904	0.892	0.855	0.832	0.843
MI	0.877	0.899	0.766	0.749	0.757
STTC	0.914	0.869	0.835	0.810	0.822
CD	0.931	0.861	0.859	0.860	0.859
HYP	0.911	0.874	0.850	0.789	0.818
AVG	0.907	0.879	0.833	0.808	0.820

**Table 11 healthcare-11-01000-t011:** Results produced by the proposed model without the Bi-GRU module on the CPSC2018 dataset.

Type	AUC	Accuracy	Precision	Recall	F1
SNR	0.964	0.939	0.768	0.843	0.804
AF	0.989	0.965	0.918	0.849	0.882
IAVB	0.982	0.975	0.922	0.831	0.874
LBBB	0.998	0.988	0.750	0.900	0.818
RBBB	0.991	0.955	0.935	0.944	0.939
PAC	0.963	0.948	0.716	0.774	0.744
PVC	0.952	0.952	0.893	0.735	0.806
STD	0.982	0.965	0.890	0.793	0.839
STE	0.961	0.964	0.571	0.545	0.558
AVG	0.976	0.961	0.818	0.802	0.807

## Data Availability

The ECG signal data used to support the findings of this study have been deposited in the PTB-XL repository (https://www.physionet.org/content/ptb-xl/1.0.3/, accessed on 11 January 2023) and CPSC 2018 (http://2018.icbeb.org/Challenge.html, accessed on 11 January 2023).

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
