# Peer review of "An ECG Classification Method Based on Multi-Task Learning and CoT Attention Mechanism"

_healthcare, 2023, doi:10.3390/healthcare11071000_

Round 1

Reviewer 1 Report

In this study, authors presented parallel task learning architecture that includes CoT attention and BI-GRU mechanism. The network’s architecture shows a slightly better performance over other model. I have several concerns before the acceptance of this paper.

1.       Could authors provide a list of abbreviations in table 1 and table 2. Why there is _ in some abbreviations such as ‘_AVB’?

2.       In this study, authors introduced auxiliary stage in the network architecture. I believe the superclass is considered as the auxiliary task in CPSC2018 data and class is considered as the auxiliary task in PTB-XL data. Could authors explain why table 3 reports the accuracy of auxiliary task (super classes) and CPSC reports the accuracy of subclasses?

3.       How is the average F1 score calculated in this study? Is it a weighted average F1 score?

Author Response

We have modified the article according to your suggestions. Please see the attachment for details.

Reviewer 2 Report

My comments are enclosed in the PDF file.

Author Response

(The authors gave the same response as above.)

Reviewer 3 Report

Dear authors

as indicated in the manuscript by highlighting the text passages in yellow read the messages and correct all the errors. Not before  you have corrected all the errors in your calculations  it will be possible to reviewe your paper in a sound manner.

Instead of mixing your results and your discussion in the section Results and Discussion (section 4 line 177 ff)  split these chapter for better understanding.

All legends have to be improved

In the discussion section please explain the relevance of using 3 decimals in your F1 score and explain why the second decimal can be used as a reliable discriminator  for a better or a poorer performance of the respective classification algorithm.

Author Response

Your suggestions have played a very important role in the revision of our article. Thank you very much for your suggestion.

Round 2

Reviewer 3 Report

Significant improvements have been made.

A major deficit remains:

1) Discussion does not adress the rather poor performance in the MI detection. Early MI detection should considerably improve the patients well being.

2) Limitations are missing

Author Response

Thank you for your suggestion. We have made changes, see the annex for details.
